# Morphological Characteristics and Transcriptome Landscapes of Chicken Follicles during Selective Development

**DOI:** 10.3390/ani12060713

**Published:** 2022-03-11

**Authors:** Ruixue Nie, Xiaotong Zheng, Wenhui Zhang, Bo Zhang, Yao Ling, Hao Zhang, Changxin Wu

**Affiliations:** 1National Engineering Laboratory for Animal Breeding, College of Animal Science and Technology, China Agricultural University, Beijing 100193, China; s20193040548@cau.edu.cn (R.N.); xiaotzheng@just.edu.cn (X.Z.); mx120170643@yzu.edu.cn (W.Z.); bozhang0606@cau.edu.cn (B.Z.); lingzi@cau.edu.cn (Y.L.); chxwu@cau.edu.cn (C.W.); 2School of Biotechnology, Jiangsu University of Science and Technology, Zhenjiang 212100, China

**Keywords:** chicken, follicle selection, granulosa cells, morphological characteristics, transcriptome sequencing

## Abstract

**Simple Summary:**

Successive follicle selection is important for egg production and reproductive performance in chickens. The molecular mechanisms of follicle selection in chickens are unclear. In the present study, the histological characteristics, reproductive hormone concentration, and transcriptional profiles of follicles were measured to identify the key genes and regulatory pathways for follicle selection. The results provide deep insights into the crucial molecular mechanism of follicle development and egg-laying performance in chickens.

**Abstract:**

Ovarian follicle selection largely depends on the transition of granulosa cells from an undifferentiated to a fully differentiated state, which is accompanied by morphological and functional changes in follicles. The processes and transcriptional regulation of follicles during follicle selection are unclear; we thus used follicles from the prehierarchal to the hierarchal stage to investigate histology, reproductive endocrinology, and transcription. The morphology of follicles changed markedly during follicle selection. The numbers of large white, small yellow, and large yellow follicles (LWF, SYF, and LYF, respectively) were 11.83 ± 2.79, 6.83 ± 2.23, and 1.00, respectively, per ovary. LYF showed thicker granulosa cell layers than those of other prehierarchal follicles. Progesterone concentrations were significantly higher in LYF than that in LWF and SYF. In total, 16,823 genes were positively expressed in LWF, SYF, and LYF. Among follicle types, 1290 differentially expressed genes were enriched regarding cell differentiation, blood vessel morphogenesis, and response to steroid hormones. Candidate genes associated with follicle selection participated in the Wnt signaling pathway, steroid hormone biosynthesis, and the TGF-β signaling pathway. We produced insights into crucial morphological characteristics of transcriptional regulation in follicle development. Our results provide an important basis for revealing the mechanism of follicle selection and potential impact on the poultry industry.

## 1. Introduction

Chicken follicle development is a remarkably complex and specifically orchestrated process that relies on the hypothalamic–pituitary axis and on paracrine and autocrine factors from ovarian follicles [1]. A sexually mature hen carries approximately 12,000 oocytes, but only a few hundred follicles are selected to mature and reach ovulation [2]. Thus, more efficient utilization of ovary follicles may help improve the egg-laying performance of poultry.

The chicken ovary undergoes a dynamic development process, and during the laying period, a series of follicles at different developmental stages occur on the ovary. In general, follicles can be divided into prehierarchal and hierarchal follicles (also termed preovulatory follicles) [3]. Prehierarchal follicles grow slowly and incorporate protein–rich yolk over several weeks [3]. During the egg–laying period, a single small yellow follicle (SYF) is recruited almost every day from a cohort of SYFs to develop into a hierarchal follicle; this process is termed follicle selection [4]. The selected follicles will develop rapidly and form a mature yolk within a few days, after which, they can ovulate and enter the oviduct [5]. Successive follicle selection is important for egg production and reproductive performance in chickens. Despite the extensive research on follicle development conducted in recent years, many questions regarding the key developmental events and the activation of follicle selection in chickens remain unanswered.

In the present study, we focused on the morphological characteristics and transcriptome of large white follicles (LWF), SYF, and large yellow follicles (LYF; also termed F5). These are three key stages of follicle selection representing follicles before selection, during selection, and after selection, respectively. The objective of our study was to identify regulatory genes involved in follicle selection in chickens. These results may provide insights for understanding the morphological changes and transcriptional regulation of follicle selection.

## 2. Materials and Methods

### 2.1. Animals and Sample Collection

A population of yellow-bearded chicken bred as hybrids of Huiyang Bearded and White Leghorn chickens was raised at the Experimental Chicken Farm of China Agricultural University (Beijing, China). All chickens were caged and reared individually in a controlled-environment house with a daily light period of 14 h. Six hens of 50 weeks of age were euthanized during the laying period and were immediately dissected to collect follicles and tissue samples. Tissues for RNA sequencing (RNA-seq) were frozen immediately in liquid nitrogen and preserved at −80 °C. All animal experiments were approved by the Beijing Municipal Committee of Animal Management and complied with the guidelines of the Ethics Committee of China Agricultural University.

### 2.2. Histological Observation of Follicles

Ovarian samples were obtained and dissected immediately, and the weight and diameter (major and minor diameter) of follicles were measured using an electronic balance (Mettler Toledo Group, Zurich, Switzerland) and Vernier caliper (Kaibao Ding Tools Co., Ltd., Wuxi, China), respectively. Follicles were fixed in 4% (*v*/*v*) buffered paraformaldehyde for 24 h, routinely desiccated, and embedded in paraffin. Sections of 5 μm thickness were cut and stained using hematoxylin and eosin. All sections were examined using a light microscope (Discover Echo, San Diego, CA, USA), and indices of the sections were calculated using ImageJ software v2.0 [6].

### 2.3. ELISA

Progesterone concentrations in follicles were measured using a Chicken Progesterone (PROG) ELISA Kit (ml059935; Shanghai Enzyme-linked Biotechnology Co., Ltd., Shanghai, China) according to the manufacturer’s instructions. Absorbance of each well was measured at 450 nm using a microplate reader (BioTek, Winooski, VT, USA). Progesterone concentrations were calculated based on the mean absorbance.

### 2.4. RNA Extraction, Library Construction, and Sequencing

Total RNA was extracted from LWF, SYF, and LYF using TRIzol reagent (Invitrogen, Carlsbad, CA, USA) following the manufacturer’s instructions. RNA quality was assessed using gel electrophoresis and spectrophotometry. cDNA library construction and high-throughput sequencing on an Illumina HiSeq 4000 platform for paired-end 150 bp sequencing platform were performed by Annoroad Gene Technology Co., Ltd. (Beijing, China). Six biological replicates were used for RNA-seq. All RNA-seq data were deposited in the NCBI Gene Expression Omnibus under the accession PRJNA795608.

### 2.5. Bioinformatics Analyses

RNA-seq reads were preprocessed using fastp v0.20.1 to remove adapters and low-base-quality sequences [7]. We mapped the clean reads to the reference chicken genome (Ensembl Gallus-gallus-6.0) using HISAT2 v2.2.1 [8]. The gene expression level was normalized based on the fragments per kilobase of transcript per million mapped reads (FPKM) values using Cufflinks v2.2.1 [9]. HTSeq v0.6.1 software was used to count the reads mapped to each gene [10]. Differentially expressed genes (DEGs) were identified using DESeq2 v1.32.0 [11]. The criteria for filtering the DEGs were $\left|\log_{2}\mathrm{fold}\;\mathrm{change}\right|$ ≥1 and *p_adj* < 0.05. Gene Ontology (GO) and Kyoto Encyclopedia of Genes and Genomes (KEGG) enrichment analysis for DEGs were performed using the clusterProfiler package [12] for R software [13], and GO terms and KEGG pathways were used for functional and biological analysis. To examine the regulation of gene expression at different developmental stages, gene expression pattern analysis was performed using the Short Time-series Expression Miner (STEM) software v1.3.8 [14]. Visualization was performed using the following tools: EVenn [15], ggplot2 [16], and GraphPad Prism v8 [17].

### 2.6. Quantitative Reverse Transcription PCR (qRT-PCR)

To verify the accuracy and repeatability of the RNA-seq results of DEGs, 10 genes were selected to measure expression levels in different follicles using qRT-PCR. The primers used for amplification of the candidate genes, including anti-Müllerian hormone (*AMH*), cyclin O (*CCNO*), cadherin 3 (*CDH3*), follistatin (*FST*), and NPC intracellular cholesterol transporter 2 (*NPC2*), were designed using the Primer-Blast tool in the National Center for Biotechnology Information (NCBI) database and are listed in Appendix A. The cDNA was synthesized in 2 µg total RNA using a FastKing gDNA Dispelling RT SuperMix (Tiangen, Beijing, China). qRT-PCR was performed on a CFX96TM Real-Time System (Bio-Rad, Hercules, CA, USA) using Real Master Mix SYBR Green I (Tiangen, Beijing, China). The total reaction volume was 20 µL, which comprised 1 µL cDNA, 10 µL 2 × Talent qPCR PreMix, 0.6 µL each of the forward and reverse primers, and 7.8 µL RNase-free water. The thermocycling program was 95 °C for 3 min, followed by 40 cycles of 95 °C for 5 s, 60 °C for 10 s, and 72 °C for 15 s. The $2^{-△△ct}$ method was used to calculate gene expression levels [18]. Expression levels of coding genes were normalized to those of *β-actin*.

### 2.7. Statistical Analyses

Data were analyzed by one-way ANOVA, followed by Duncan’s test using *SPSS* software v25 (SPSS Inc., Chicago, IL, USA). Data are presented as means ± standard error (SE). Significance is reported at *p* < 0.05.

## 3. Results

### 3.1. Number and Size of Follicles at Development Stage

The average weight of a complete chicken ovary was 46.26 ± 9.08 g. In an ovary, there are multiple prehierarchal follicles at each developing stage and one hierarchal follicle at each stage, and the follicle color gradually changes from white to yellow (Figure 1). The numbers of LWF, SYF, and LYF in laying hens were 11.83 ± 2.79, 6.83 ± 2.23, and 1.00, respectively (Table 1 and Figure 2A–C). The major, minor, and average diameters of LYF were significantly higher than those of LWF and SYF (Table 1 and Figure 2D). The weight of LWF and SYF was 0.07 ± 0.02 g and 0.16 ± 0.06 g, respectively, which was significantly lower than that of LYF (0.93 ± 0.19 g; Table 1 and Figure 2E).

### 3.2. Histological Observation and Follicle Progesterone Concentrations

There were only one or two granulosa layers in LWF and SYF, and LYF had multiple layers (Figure 3A–C). During follicle development, the granulosa layer of LYF (15.67 ± 2.61 μm) was significantly thicker than those of LWF (7.53 ± 1.58 μm) and SYF (9.92 ± 1.20 μm; Figure 3D). The area of single granulosa cells (GCs) increased significantly (Figure 3E), and the GCs were arranged more loosely in LYF. Blood vessels were observed in SYF and LYF sections (Figure 3B,C), which may be because it is convenient to provide more nutrients to promote follicle development and stimulate follicle selection. Progesterone is a steroid hormone secreted by GCs [19], and its concentration in the follicles was measured. There was a significant difference in the concentration of progesterone among the three stages of follicles (Figure 3F). After follicle selection, the progesterone concentration of LYF was significantly higher than that of prehierarchal follicles. The genes involved in progesterone synthesis were also significantly upregulated in LYF (Figure 3G–I).

### 3.3. Experimental Design and RNA-Seq Overview

We collected tissues of LWF, SYF, and LYF from hens to examine the respective transcriptomes (Figure 4A). The concentration, purity, and other indexes of RNA obtained from follicles conformed to the sequencing requirements (Appendix A). In total, 1163.90 million raw reads were generated from 18 RNA-seq libraries (Appendix A). Of these reads, an average of 93% were multiple-mapped and 85.27%–88.05% uniquely mapped to the reference genome (Appendix A). In the gene annotation files of uniquely aligned sequences, there were 73.80 ± 0.42%, 16.10 ± 0.18%, and 10.10 ± 0.36% reads of each sample that aligned to mRNA, intergenic, and intron regions, respectively (Appendix A). A total of 16,791, 16,823, and 16,062 genes were identified in LWF, SYF, and LYF, respectively (Appendix A). There were 74 DEGs in LWF vs. SYF, 77 in SYF vs. LYF, and 1287 in LWF vs. LYF, of which 12 occurred in all three comparisons (Figure 4B and Appendix A). The DEGs in LWF vs. LYF included almost all DEGs of the other two comparisons (Figure 4B).

### 3.4. Validation of RNA-Seq Data by qRT-PCR

To validate the RNA-seq results, 10 genes were selected and quantified using qRT-PCR. The genes and the respective primer sequences are shown in Appendix A. The expression of genes showed similar trends as the RNA-seq data (Figure 5A). Linear regression analysis of the $\log_{2}\mathrm{fold}\;\mathrm{change}$ in gene expression between RNA-seq and qRT-PCR showed a significant positive correlation ($R^{2}$ = 0.91), confirming that the RNA-seq results were reliable (Figure 5B).

### 3.5. Functions of DEGs between LWF and SYF

A total of 74 DEGs were identified by comparing LWF with SYF, which included 68 upregulated and 6 downregulated genes (Figure 6A and Appendix A). Of the DEGs, transmembrane protein 72 (*TMEM72*), solute carrier family 5 member 5 (*SLC5A5*), inhibin alpha subunit (*INHA*), *CDH3*, *CCNO*, and *AMH* were highly expressed in both LWF and SYF with high fold change values ($\log_{2}\mathrm{fold}\;\mathrm{change}$ ranged from −1.1 to 5.3). These DEGs were mainly enriched in GO terms of cell differentiation, cell–cell adhesion mediated by cadherin, steroid hormone receptor activity, cellular developmental process, and SMAD protein signal transduction (Figure 6B; comprehensive list in Appendix A) and KEGG pathways of cytokine–cytokine receptor interaction, transforming growth factor (TGF)-β signaling pathway, Wnt signaling pathway, and glycerophospholipid metabolism (Figure 6C; comprehensive list in Appendix A).

### 3.6. Functions of DEGs between SYF and LYF

A total of 77 DEGs were detected in SYF vs. LYF, with 44 upregulated and 33 downregulated genes (Figure 7A and Appendix A). Expressions of actin alpha cardiac muscle 1 (*ACTC1*), nuclear receptor subfamily 5 group A member 2 (*NR5A2*), and *NPC2* were upregulated during this period with six-fold change, and leukocyte cell-derived chemotaxin 1 (*LECT1*) and *LOC422926* showed low expression in LYF. DEGs in the SYF vs. LYF comparison were mainly enriched regarding steroid hormone-mediated signaling pathway, blood vessel morphogenesis, transforming growth factor beta production, response to steroid hormone, and steroid hormone receptor activity (Figure 7B; comprehensive list in Appendix A). KEGG pathway analysis of the DEGs between SYF and LYF included the following processes: ether lipid metabolism, glycerolipid metabolism, and PPAR signaling pathway (Figure 7C; comprehensive list in Appendix A).

### 3.7. Functions of DEGs between LWF and LYF

The number of DEGs between LWF and LYF was large (n = 1287; Appendix A). The DEGs between LWF and LYF included 619 upregulated and 668 downregulated genes (Figure 8A). The majority of the DEGs in LWF vs. SYF and SYF vs. LYF were included in LWF vs. LYF. *TMEM72*, GDNF family receptor alpha 3 (*GFRA3*), *SLC5A5*, stimulated by retinoic acid 6 (*STRA6*), *INHA*, and *CDH3* are the common elements of LWF vs. SYF and LWF vs. LYF. In addition, 62 coexpressed DEGs of SYF vs. LYF and LWF vs. LYF included *ACTC1*, albumin (*ALB*), *NPC2*, *LECT1*, and *LOC422926.* Moreover, there were 12 coexpressed DEGs, including placental growth factor (*PGF*), *NR5A2*, galactosylceramidase (*GALC*), phospholipase C gamma 1 (*PLCG1*), matrilin 3 (*MATN3*), and *AMH*. GO enrichment analysis of DEGs indicated cell differentiation, blood vessel development, extracellular region, anatomical structure morphogenesis, and lipid transport (Figure 8B; comprehensive list in Appendix A). KEGG analysis of functional pathways of LWF vs. LYF produced TGF-β signaling pathway, ECM-receptor interaction, steroid hormone biosynthesis, retinol metabolism, tight junction, and biosynthesis of amino acids (Figure 8C; comprehensive list in Appendix A).

### 3.8. Gene Expression Dynamics and Transcriptional Profiles around Follicle Selection

We examined DEGs around the stages of follicle selection using the Time-series Expression Miner (STEM) software. A total of 1290 DEGs clustered in 16 profiles, of which 6 were significant (*p* < 0.05), including 3 downregulated patterns (profiles 2, 3, and 7) and 3 upregulated patterns (profiles 12, 13, and 15) (Figure 9). Profiles 2, 3, and 7 contained 290, 126, and 184 DEGs, respectively, whereas profiles 12, 13, and 15 contained 156, 109, and 137 DEGs, respectively (Appendix A). The 3 upregulated and 3 downregulated clusters were merged into a new cluster that was subjected to KEGG pathway enrichment analysis in subsequence.

### 3.9. Key Signaling Pathways in Follicle Selection

To investigate the regulatory network in follicle selection, we analyzed the expression of components of key signaling pathways, including the Wnt signaling pathway, steroid hormone biosynthesis, and TGF-β signaling pathway. The levels of the ligand *WNT4* of the Wnt signaling pathway were upregulated during follicle selection. The receptor *FZD4* and the downstream target gene *NFATC1* were upregulated from LWF to LYF. *WIF1* and *SFRP2* are inhibitory factors of the Wnt signaling pathway and showed a pattern of downregulation (Figure 10A). These findings suggest that the Wnt signaling pathway is involved in the control of follicular development.

Key DEGs of the steroid hormone biosynthesis pathway are summarized in Figure 10B. Cholesterol is a substrate for steroid biosynthesis. Steroidogenic acute regulatory protein (*StAR*) transports cholesterol molecules from the outer mitochondrial membrane to the inner mitochondrial membrane [20]. The cholesterol side-chain cleavage enzyme P450scc (encoded by the *CYP11A1* gene) is located on the inner mitochondrial membrane, which converts cholesterol to pregnenolone. Pregnenolone is catalyzed by *HSD3B1* to produce progesterone. Under the joint action of key genes, such as *CYP17A1*, *HSD3B1*, and *CYP19A1*, pregnenolone is converted to estrone. The DEGs related to progesterone biosynthesis were significantly upregulated during follicle selection.

We analyzed the expression pattern of significant members of the TGF-β signaling pathway during follicle selection. The TGF-β superfamily is the largest signal protein family, and more than 40 members have been identified to date. The expression of *AMH* and its receptor *AMHR2* was significantly reduced from LWF to LYF. In contrast, the expression of *GDF11*, *INHA*, *INHBA*, and their receptor, *ACVR2A*, gradually increased during follicle selection. Other members of the TGF-β superfamily, *BMP4* and *GDF7*, were also upregulated throughout follicular development, and their receptor, *BMPR2*, was also upregulated (Figure 10C). This observation suggests that TGF-β superfamily members may play different roles during the development of chicken follicles, and correspondingly, the downstream signaling effectors *SMAD5* and *SMAD2Z* and the target genes *ID2* and *ID3* were upregulated.

## 4. Discussion

Follicle selection is a widespread phenomenon in vertebrates [1] that can influence egg production performance in laying hens. However, the cellular mechanisms underlying follicle selection are unclear, and they should be elucidated to improve the laying performance of chickens. Recently, studies have investigated chicken follicle selection [21,22]. *Wnt4* can increase *StAR*, *CYP11A1*, and *FSHR* expression; promote GC proliferation of prehierarchal and hierarchal follicles; and ultimately drive follicle selection [21]. A total of 855 DEGs and 259 differentially expressed proteins were identified by transcriptomic and proteomic comparative analyses of SFY and F6, which may contribute to chicken follicle selection and development [22]. In the current study, follicles in the transition from LWF to LYF showed different morphological characteristics and transcriptomes. This is the first study on the development of follicles of the chicken ovary using section observation and high-throughput transcriptome technology to uncover the mechanisms of follicle selection.

In the chicken ovary, a single prehierarchal follicle is selected to mature into a hierarchical line from a SYF cohort. With the development of follicles, the number of follicles in the stage decreases. Only five or six hierarchal follicles were identified after follicle selection. In this study, there were approximately 12 LWF, 7 SYF, and 1 LYF on each hen’s ovary, with weights of 0.07, 0.16, and 0.93 g per follicle, respectively. Follicular transition occurs through the surrounding somatic cells, functional differentiation of GC, and incorporation of the theca layer [23]. We produced paraffin sections of the chicken follicular wall to observe their morphological characteristics. The results revealed that the granulosa layer was significantly thicker in LYF than in LWF and SYF during follicle selection. The increased thickness of the granulosa layer is due to proliferation of GCs, which promotes transition from prehierarchal to hierarchal follicles [3,24]. With the development of the GC layer, the area of a single GC gradually increased, and the arrangement showed a loose trend. By observing the sections of the follicle wall, we found that there were blood vessels surrounding the granulosa layer, which developed after selection. The most recently selected follicle, an LYF, begins to take up large amounts of yolk and grows rapidly over a period of days [25,26]. The changes in morphological characteristics are the physiological basis for the realization of the biological processes described above: the extensive vascular system and faster blood flow efficiently facilitate higher uptake of lipid-rich yolk, and the looser GC arrangement can facilitate the transport of macromolecular substances.

Hundreds of DEGs and dozens of important signaling pathways were revealed in this study, including the Wnt signaling pathway, TGF-β signaling pathway, and steroid hormone biosynthesis. Previous studies suggested that the Wnt signaling pathway is involved in the control of follicular development in humans, rats, mice, and pigs, working together with follicle-stimulating hormone (FSH) [27,28,29,30]. In chickens, GCs from hierarchal follicles (Hie-GCs) respond to FSH treatment via increased cyclic adenosine monophosphate (cAMP) production, but GCs from prehierarchal follicles (pre-GCs) could not [3]. There may exist an inhibitory or nonactivated signal resulting in pre-GCs not responding to FSH [3]. *Wnt4* can stimulate follicle selection by enhancing *FSHR* expression and GC proliferation [21]. *Wnt4* is known to regulate differentiation of the embryonic ovary and differentiation of GCs from both prehierarchal and hierarchal follicles [21,31]. In our study, *Wnt4* was significantly upregulated from LWF to LYF, which means that *Wnt4* is likely to drive the prehierarchal-to-hierarchal stage transition by promoting the development of the GC layer, suggesting that *Wnt4* may also be involved in follicle histological characteristic changes during follicle selection, including an increase in the thickness of the GC layer and an increase in the area of each GC. Our results also showed that *FZD4* receptors and the downstream target gene *NFATC1* were highly expressed in LYF. *FZD4*, a key receptor of the Wnt signaling pathway that has been previously described in mice, has been shown to upregulate vascular growth and organization [32,33]. *NFATC1* is a member of the nuclear factor of activated T cell (NFAT) family and is upregulated in various cancers and mediates cell growth, migration, and invasion [34,35,36]. This indicates that the Wnt signaling pathway is activated in the selected follicle, LYF, and is involved in promoting vascular formation and GC proliferation, thus driving the follicle selection mechanism.

In the current study, we analyzed the expression of key components of the TGF-β signaling pathway. Consistent with previous findings in poultry, this signaling pathway was involved in initiating follicle selection [37,38,39]. *AMH* is a member of the TGF-β family located in GCs, and its levels decline significantly during follicle development [39,40]. *AMH* binding to its specific primary receptor, *AMHR2*, in chicken inhibits the growth of prehierarchal follicles and prevents premature maturation of follicles in laying hens [40]. Therefore, *AMH* plays a role in maintaining quiescence by inhibiting the transition from prehierarchy to hierarchy [41]. *GDF11*, *INHA*, *BMP4*, and *GDF7* bind to *ACVR2A* and *BMPR2* receptors, respectively, and play vital roles in preventing GC differentiation prior to follicle selection and folliculogenesis [38,41]. *SMADs* are downstream intracellular signaling molecules of the TGF-β ligands, and knockout key members of *SMADs* are downstream intracellular signaling molecules of the TGF-β ligands; knocking out key members of *SMADs* impairs primordial germ cell development in mice [42]. Consistent with previous findings in other animal models [43,44], TGF-β signaling pathway ligands and their targets *ID2* and *ID3* were highly expressed in chicken LYF. The presence of this gene and its target in this pathway led us to assume that the TGF-β signaling pathway is an important factor driving follicle selection.

GCs of hierarchal follicles begin to produce progesterone, possibly under the stimulatory regulation of FSH [45]. Progesterone is a common steroidal hormone that plays an important regulatory role in the female reproductive system [46]. The steroidogenic competency transition corresponds to the time of follicle selection, suggesting that progesterone synthesis is related to follicle selection [47]. In the current study, the progesterone concentration of LYF was significantly higher than that of LWF and SYF. The principal pathways of steroidogenesis were elucidated [20], and the steroid hormone biosynthesis pathway was also enriched in the RNA-seq data. The steroid hormone biosynthesis pathway plays a role in follicle development in many species; in Leizhou black ducks, the steroid biosynthesis pathway affects egg production differences as assessed through transcriptome sequencing [48]. The steroid biosynthetic process also positively regulates hormone secretion during goose GC development induced by FSH [49].

*StAR* is a member of the steroid hormone biosynthesis pathway, and the *StAR* protein regulates the rate-limiting step in steroidogenesis and the transport of cholesterol from the outer to the inner mitochondrial membrane [50,51]. The increase in steroid production is mediated by the cAMP-dependent pathway and *StAR* gene rapid transcription, which is stimulated by gonadotropic hormones (e.g., LH or FSH) [52,53]. Although the exact mechanisms that promote *StAR* transcription in chickens remain unclear, numerous studies reported respective transcription factors in humans and rodents, such as steroidogenic factor 1 (*SF-1*) [54], CCAAT/enhancer binding proteins (C/EBPs) [55], *GATA-4* [56], and *DAX-1* [51,57,58]. Similar to several other steroidogenic enzyme genes, the *StAR* promoter lacks cAMP-responsive elements [52], and cAMP responsiveness of the *StAR* promoter is conferred by the transcription factors mentioned above. Gonadotropin-induced *StAR* expression is linked to the significant production of progesterone, and the initial differentiation of GCs during follicle development in poultry has been confirmed in many studies [21,59,60]. Our results revealed that the expression patterns of *StAR* were markedly upregulated during the entire follicle selection period. Therefore, *StAR* was substantially affected by upstream functional genes within the transition of prehierarchal to hierarchal follicles.

## 5. Conclusions

We describe the histological characteristics and corresponding transcriptome profiles of LWF, SYF, and LYF, which represent stages around follicle selection. The results establish a foundation for further investigation of follicle selection mechanisms by elucidating global functional genes and important signaling pathways involved in ovarian follicular development. Moreover, this study generated a considerable resource for poultry breeding to enhance egg-laying performance.

## Figures and Tables

**Figure 1 animals-12-00713-f001:**
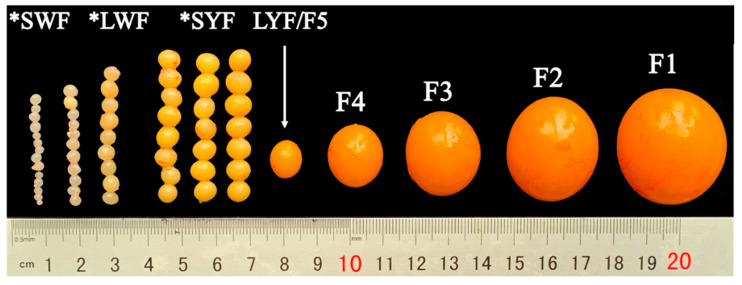
Ovarian follicles of laying hens. Asterisks indicate prehierarchal follicles; SWF: small white follicle; LWF: large white follicle; SYF: small yellow follicle; F5–F1 represent hierarchal follicles, which are sorted from smallest to largest in diameter. F5 represents the most recently selected follicle (LYF: large yellow follicle).

**Figure 2 animals-12-00713-f002:**
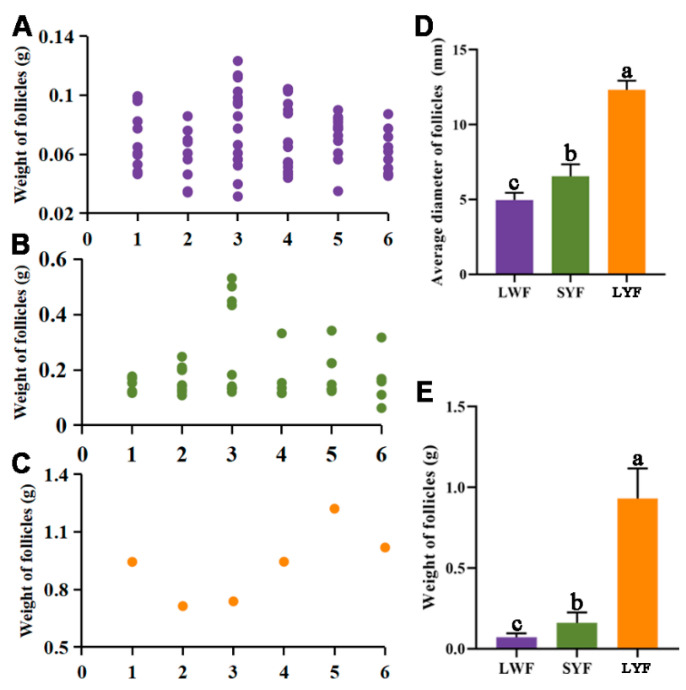
Number and size of follicles at development stages. (**A**–**C**) Diameter and weight of large white follicles (LWF) (**A**), small yellow follicles (SYF) (**B**), and large yellow follicles (LYF) (**C**). Each number on the *x*-axis represents an individual hen, and each dot represents one follicle. (**D**,**E**) Average diameter (**D**) and weight (**E**) of follicles around follicle selection. Data represent the mean ± standard error (*n* = 6), and different letters indicate significant difference (*p* < 0.05).

**Figure 3 animals-12-00713-f003:**
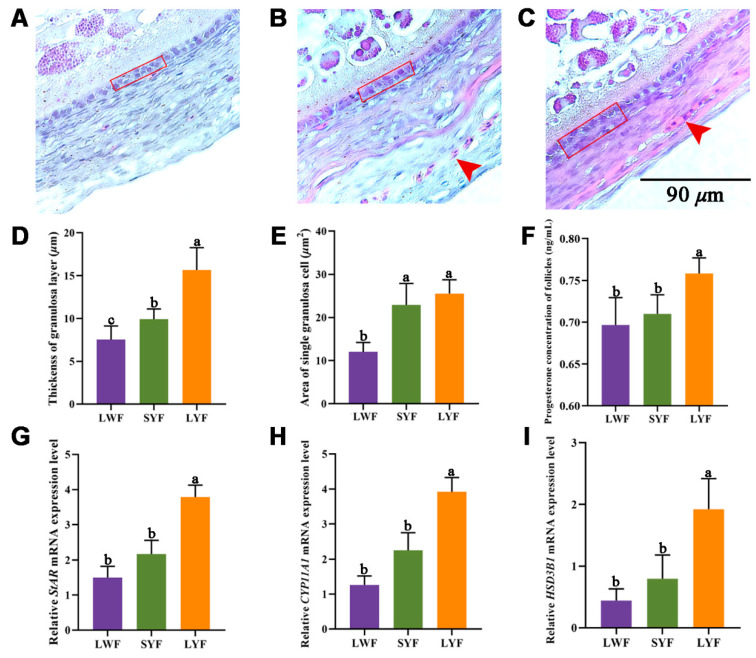
Histological observation and progesterone concentration of follicles. (**A**–**C**) Paraffin sections of large white follicles (LWF) (**A**), small yellow follicles (SYF) (**B**), and large yellow follicles (LYF) (**C**). Sections of 5 μm thickness were cut and stained with hematoxylin and eosin. The red rectangles indicate the granulosa cell layer, and the red arrows indicate the blood vessels; scale bar = 90 μm. (**D**) The thickness of granulosa cell layers. (**E**) The area of single granulosa cells. Indexes were calculated using ImageJ software. (**F**) Concentration of progesterone in follicles around follicle selection. (**G**–**I**) Relative mRNA expression of key genes involved in progesterone synthesis: *StAR* (**G**) *CYP11A1* (**H**), and *HSD3B1* (**I**). Values are as mean ± standard error (*n* = 6); different letters indicate significant difference (*p* < 0.05).

**Figure 4 animals-12-00713-f004:**
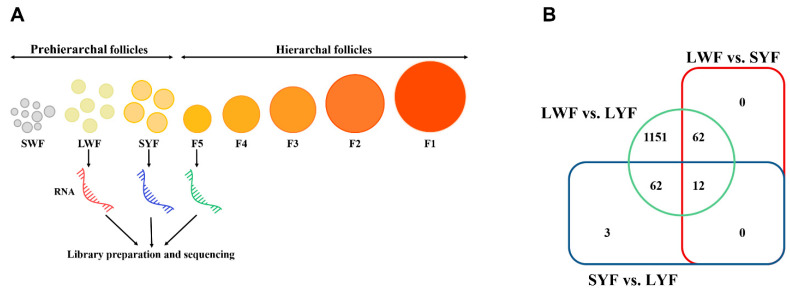
Experimental design of RNA-seq and differentially expressed gene (DEG) distribution. (**A**) Schematic illustration of the study workflow. Large white follicles (LWF), small yellow follicles (SYF), and large yellow follicles (F5/LYF) were collected for total RNA extraction and subjected to RNA-seq. (**B**) Venn diagram of DEGs; the criteria for filtering the DEGs were *p_adj* < 0.05 and $\left|\log_{2}\mathrm{fold}\;\mathrm{change}\right|$ ≥ 1.

**Figure 5 animals-12-00713-f005:**
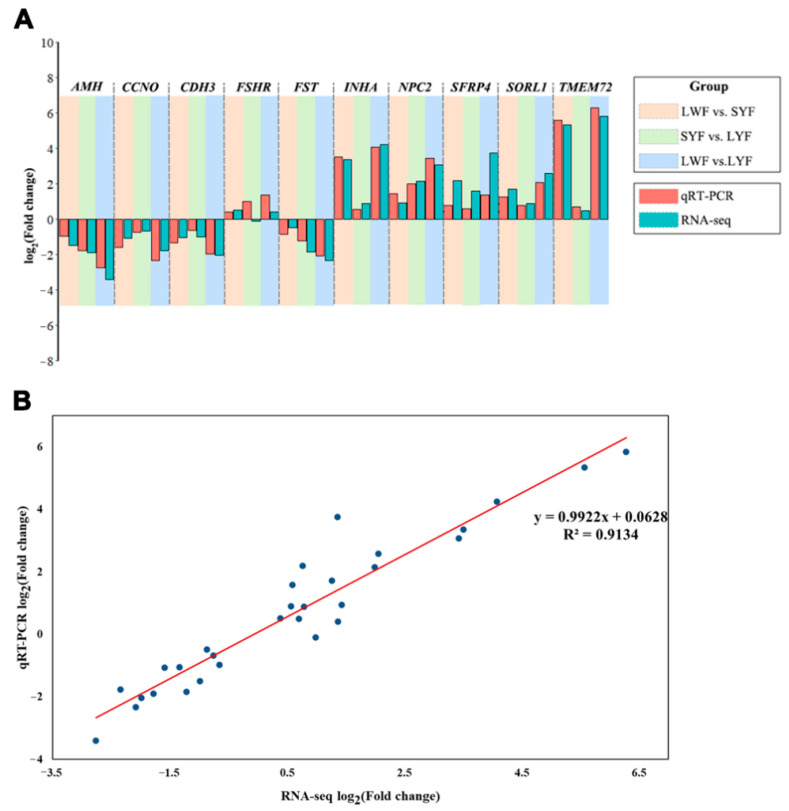
Validating the RNA-seq data by qRT-PCR. (**A**) Comparison of $\log_{2}\mathrm{fold}\;\mathrm{change}$ in 10 differentially expressed genes (DEGs) between qRT-PCR and RNA-seq and (**B**) regression analysis of $\left|\log_{2}\mathrm{fold}\;\mathrm{change}\right|$ values between qRT-PCR and RNA-seq. A high $R^{2}\;$ indicated that the RNA-seq data were considered to be of a high accuracy.

**Figure 6 animals-12-00713-f006:**
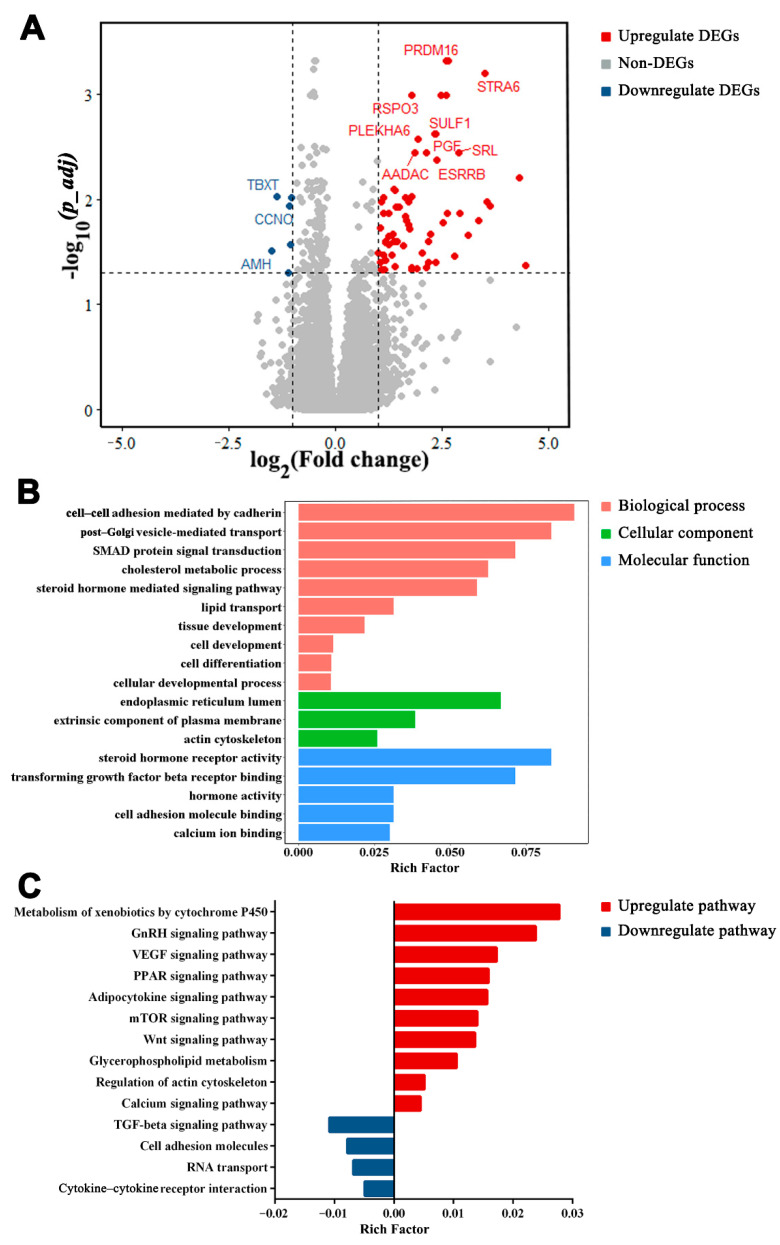
Differentially expressed genes (DEGs) in large white follicles (LWF) vs. small yellow follicles (SYF). (**A**) Volcano map of DEGs in LWF vs. SYF. DEGs with *p_adj* < 0.05 and $\log_{2}\mathrm{fold}\;\mathrm{change}$ ≥ 1 are marked in red, and DEGs with *p_adj* < 0.05 and $\log_{2}\mathrm{fold}\;\mathrm{change}$ ≤ −1 are marked in blue. (**B**) Gene Ontology functional classification of DEGs in biological process, cellular component, and molecular function categories. (**C**) Kyoto Encyclopedia of Genes and Genomes pathways of DEGs.

**Figure 7 animals-12-00713-f007:**
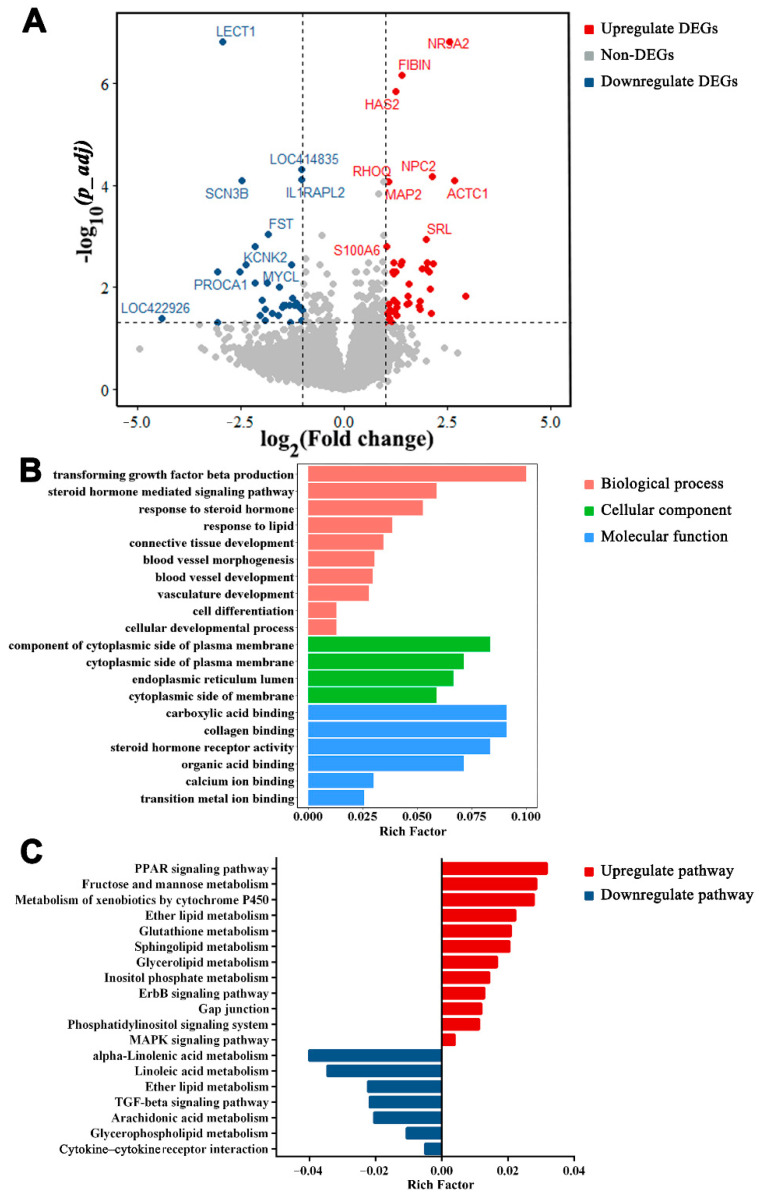
Differentially expressed genes (DEGs) in small yellow follicles (SYF) vs. large yellow follicles (LYF). (**A**) Volcano map of DEGs in SYF vs. LYF. DEGs with *p_adj* < 0.05 and $\log_{2}\mathrm{fold}\;\mathrm{change}$ ≥ 1 are marked in red, and DEGs with *p_adj <* 0.05 and $\log_{2}\mathrm{fold}\;\mathrm{change}$ ≤ −1 are marked in blue. (**B**) Gene Ontology functional classification of DEGs in biological process, cellular component, and molecular function categories. (**C**) Kyoto Encyclopedia of Genes and Genomes pathways of DEGs.

**Figure 8 animals-12-00713-f008:**
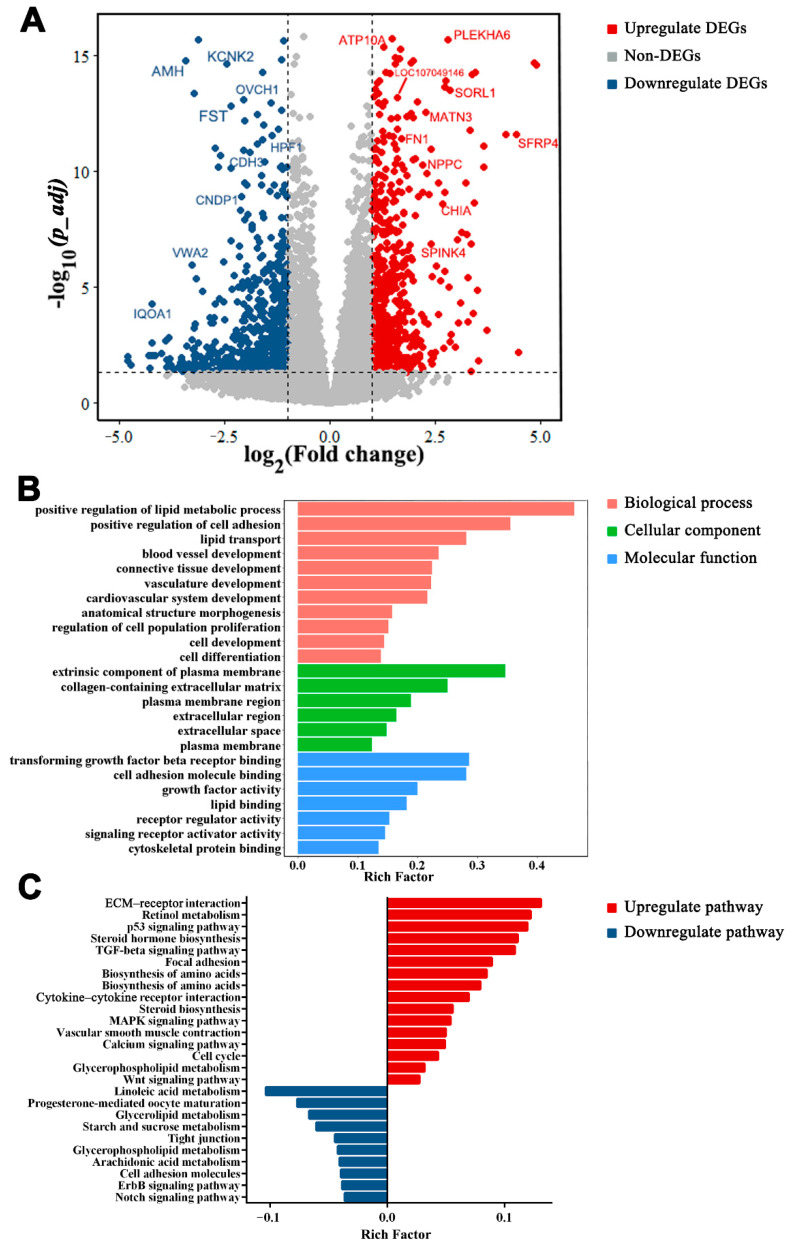
Differentially expressed genes (DEGs) in large white follicles (LWF) vs. large yellow follicles (LYF). (**A**) Volcano map of DEGs in LWF vs. LYF. DEGs with *p_adj* < 0.05 and $\log_{2}\mathrm{fold}\;\mathrm{change}$ ≥ 1 are marked in red, and DEGs with *p_adj* < 0.05 and $\log_{2}\mathrm{fold}\;\mathrm{change}$ ≤ −1 are marked in blue. (**B**) Gene Ontology functional classification of DEGs in biological process, cellular component, and molecular function categories. (**C**) Kyoto Encyclopedia of Genes and Genomes pathways of DEGs.

**Figure 9 animals-12-00713-f009:**
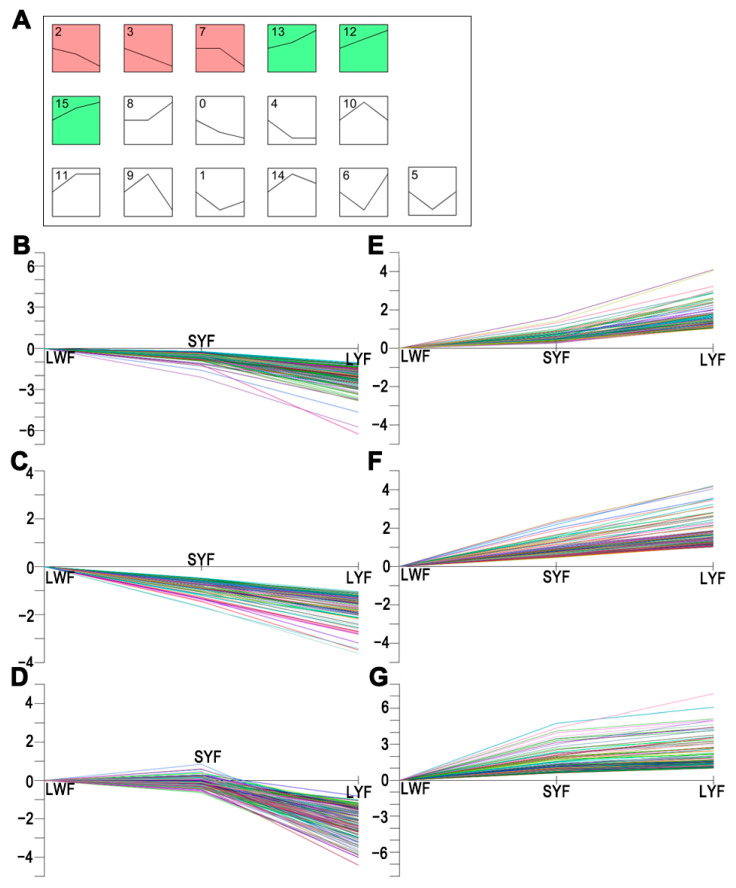
Gene expression dynamics profiles around follicle selection. (**A**) All mRNA expression profiles. All mRNAs could be clustered into 16 profiles, of which 6 were significant (*p* < 0.05). (**B**–**D**) Three downregulated patterns (profiles 2, 3, and 7, respectively). (**E**–**G**) Three upregulated patterns (profiles 13, 12, and 15, respectively).

**Figure 10 animals-12-00713-f010:**
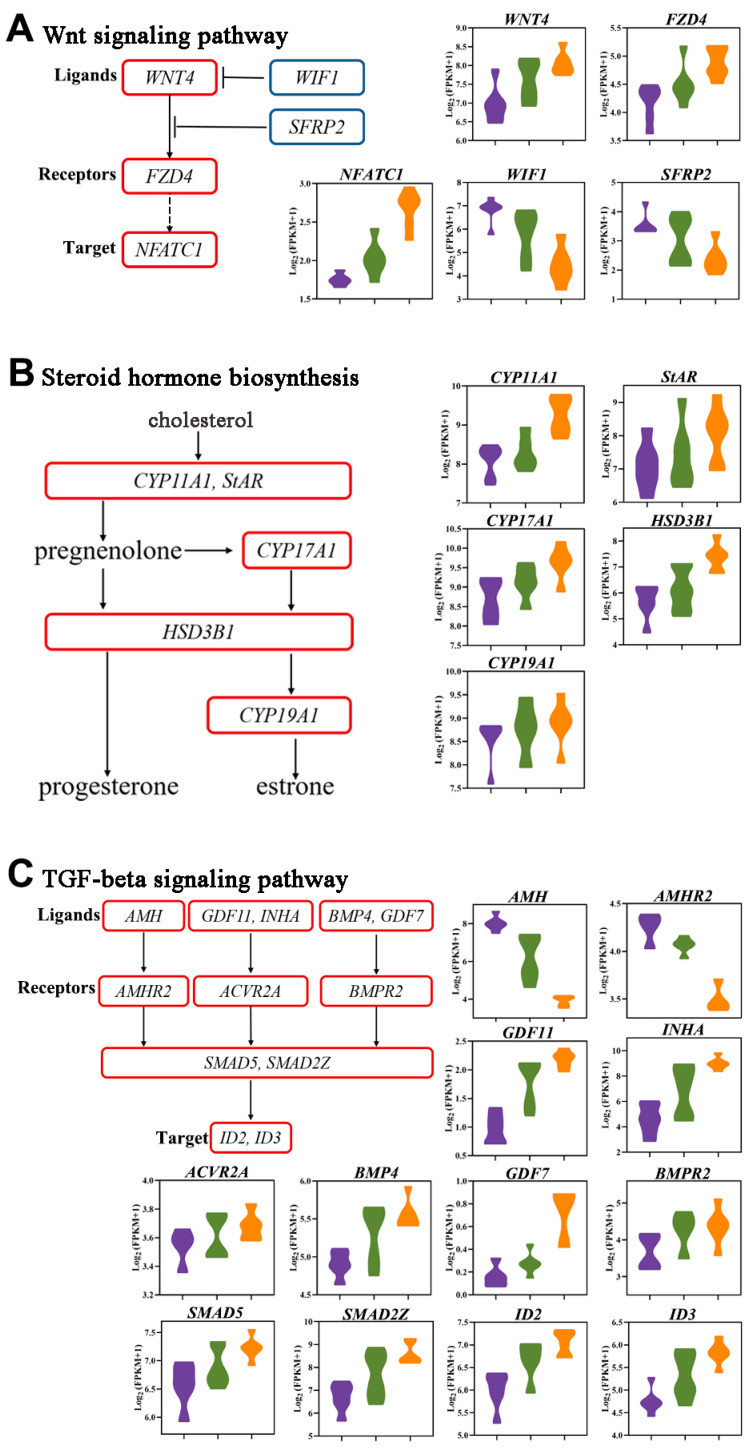
Signaling pathways involved in follicle selection. (**A**) Wnt signaling pathway. (**B**) Steroid hormone biosynthesis. (**C**) TGF-β signaling pathway. Violin plots show the relative expression levels, $\log_{2}\left[\mathrm{FPKM}+1\right],$ of the ligands and receptors and target differentially expressed genes (DEGs) in each Kyoto Encyclopedia of Genes and Genomes pathway. The purple, green, and orange violins represent LWF, SYF, and LYF, respectively. Flow diagrams show the relationship among these genes.

**Table 1 animals-12-00713-t001:** Parameters of follicles around follicle selection.

	LWF	SYF	LYF	*p*-Value
Quantity (number) n ^1^	11.83 ± 2.79 ^a^	6.83 ± 2.23 ^b^	1 ^c^	<0.001
Weight (g) ^2^	0.07 ± 0.02 ^c^	0.16 ± 0.06 ^b^	0.93 ± 0.19 ^a^	<0.001
Major diameter (mm) ^3^	5.23 ± 0.53 ^c^	6.92 ± 0.91 ^b^	13.02 ± 0.61 ^a^	<0.001
Minor diameter (mm) ^4^	4.71 ± 0.49 ^c^	6.13 ± 0.81 ^b^	11.61 ± 0.84 ^a^	<0.001
Average diameter (mm) ^5^	4.97 ± 0.49 ^c^	6.55 ± 0.81 ^b^	12.31 ± 0.67 ^a^	<0.001

LWF = large white follicles; SYF = small yellow follicles; LYF = large yellow follicle. ^1^ Numbers of follicles around follicle selection, including LWF, SYF, and LYF. ^2^ Weight of LWF, SYF, and LYF. ^3,4,5^ Maximum, minimum, and average diameters of follicles, respectively. The values in the table are compared within each row, and different letters indicate significant difference (*p* < 0.05).

## Data Availability

The original data in this study are openly available in the Dataverse Project, reference number PRJNA795608.

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
