# Peer review of "Morphological Characteristics and Transcriptome Landscapes of Chicken Follicles during Selective Development"

_animals, 2022, doi:10.3390/ani12060713_

Round 1

Reviewer 1 Report

See attached. 

Reviewer 2 Report

Morphological characteristics and transcriptome landscapes of chicken follicles during selective development

Comments

The authors report the different genes and signaling pathways involved in follicle selection for ovulation in chicken. The manuscript is well-written and will be useful for reproductive physiologists and poultry scientists.

Abstract

Include a sentence on the potential impact of this study on the poultry industry.

Results

Make sure to expand all abbreviations the first time they appear in the text, especially gene names.

Table1: In the footnote, mention how to compare the results, whether it is within a column or row. Also, for quantity (number), if the means of only LWF and SYF were compared, include that too.

Figure 3: In the footnote, correct HSD3β1.

Figure 4: In the footnote, expand DEGs.

Figure 5A: The colors representing groups are too light.

Figure 10: In the relative expression chart, what does each color represent?

Reviewer 3 Report

Comments to the Authors

This paper by Nie et al gives the histological characteristics, reproductive hormone concentration, and transcriptional profiles of follicles in order to identify the key genes and regulatory pathways for follicle selection. The results provide deep insights into the crucial molecular mechanism of follicle development and egg-laying performance in chickens.

In order to study regulatory genes involved in the follicle selection in chickens, the authors focused on morphological characteristics and the transcriptome of large white follicles (LWF), SYF and large yellow follicles (LYF) that represent three key stages of follicle selection: follicles before selection, during selection, and mostly after selection, respectively.

The paper is well-written, the experiments appear to be carried out competently. The topic is interesting and the manuscript has potential for publication nevertheless, before I have a few minor revisions listed below:

line 120: the authors use 1 µl of cDNA but how much the concentration of cDNA for each sample?

Paragraph 2.6. the text lacking details about the procedure of retrotranscription procedure

Table 1: Why are the LYF follicles not considered during the statistical analysis? I understand the only one LYF is present for each hen but the authors should use one non-parametric test.

In Figure 3 the authors should invert the order of representation and follow the order in which the types of follicles are descriptive in the text.

In Figure 3 A-C: The granulosa cell layers of LWF and SYF appear thicker than that of LYF but the text and data (Figure 3D) show the contrary. In my opinion, a more representative picture should be inserted

Round 2

Reviewer 1 Report

Sufficient improvements were made based on the provided comments.